# Identifying and Characterizing Candidate Genes Contributing to a Grain Yield QTL in Wheat

**DOI:** 10.3390/plants13010026

**Published:** 2023-12-20

**Authors:** Md Atik Us Saieed, Yun Zhao, Shahidul Islam, Wujun Ma

**Affiliations:** 1Food Futures Institute, School of Health, Education & Environment, Murdoch University, Perth, WA 6150, Australia; atik.sst@bau.edu.bd (M.A.U.S.); y.zhao@murdoch.edu.au (Y.Z.); shahidul.islam.1@ndsu.edu (S.I.); 2Department of Seed Science & Technology, Bangladesh Agricultural University, Mymensingh 2202, Bangladesh; 3Department of Plant Sciences, North Dakota State University, Fargo, ND 58108, USA; 4College of Agronomy, Qingdao Agriculture University, Qingdao 266109, China

**Keywords:** quantitative trait locus (QTL), bioinformatics, gene identification

## Abstract

The current study focuses on identifying the candidate genes of a grain yield QTL from a double haploid population, Westonia × Kauz. The QTL region spans 20 Mbp on the IWGSC whole-genome sequence flank with 90K SNP markers. The IWGSC gene annotation revealed 16 high-confidence genes and 41 low-confidence genes. Bioinformatic approaches, including functional gene annotation, ontology investigation, pathway exploration, and gene network study using publicly available gene expression data, enabled the short-listing of four genes for further confirmation. Complete sequencing of those four genes demonstrated that only two genes are polymorphic between the parental cultivars, which are the *ferredoxin*-like protein gene and the *tetratricopeptide*-repeat (*TPR*) protein gene. The two genes were selected for downstream investigation. Two SNP variations were observed in the exon for both genes, with one SNP resulting in changes in amino acid sequence. qPCR-based gene expression showed that both genes were highly expressed in the high-yielding double haploid lines along with the parental cultivar Westonia. In contrast, their expression was significantly lower in the low-yielding lines in the other parent. It can be concluded that these two genes are the contributing genes to the grain yield QTL.

## 1. Introduction

Grain yield is the most complex trait in wheat and is the prime target of wheat breeding. However, yield is a cumulative reflection of numerous biological processes related to plant growth and development and their interaction with micro-environments. As such, the improvement of grain yield becomes one of the most challenging tasks in wheat breeding [1].

The quantitative trait locus (QTL) has been utilized in the breeding program through marker-assisted selection (MAS) to improve yield. QTL mapping allows identifying the specific chromosomal location and evaluating their effects [2,3]. Plant QTL mapping accomplishes considerable progress, along with the identification of several grain yield and protein content-related QTLs. However, only a few QTLs have been successfully adopted for MAS [4,5]. This is perhaps because the genetic gain of grain yield has not been achieved on a large scale. The reason might be that this approach is usually not gene-specific but rather chromosomal location-specific, which usually harbors many genes. Another major bottleneck is that most of the QTLs were not validated, as this requires a considerable amount of effort and investment [6]. Identifying the contributing gene of a QTL will allow setting a precise genetic target in the breeding program.

However, the approach of functional confirmation of all genes in the region is challenging [7] due to the requirements of excessive resources and time. An alternative might be the conventional way of gene identification, known as positional cloning, which has been a common way to allocate the shortest genetic interval, known as fine mapping [8]. It improves the map resolutions through a larger mapping population and more molecular markers. Therefore, identifying candidate genes via positional cloning is also time-consuming, labor-intensive, and expensive. In contrast, the computational approach of prioritizing QTL candidate genes will help overcome these pitfalls [9].

The computational approach, generally called the in silico approach, employs several steps to screen out the non-contributing genes, which include identifying variation in the DNA sequence and SNP (single nucleotide polymorphism), characterizing biological pathways, and understanding functional evolution. It focuses on the integration and visualization of all the available information from different platforms. In silico approaches to short-listing the candidate genes provide the opportunity to reduce the workload of downstream laboratory experiments and the costs involved [10]. The recently released whole genome sequence of bread wheat (*Triticum aestivum*) can undoubtedly assist in overcoming the bottleneck. Also, significant advancements in producing powerful tools for in silico analysis in recent years have made it possible to follow this process.

Although the in silico approach provides a precise list of candidate genes, downstream laboratory activities are still required for functional confirmation. This, in particular, is making the laboratory tasks manageable and cost-effective. One of the most logical laboratory approaches to support the computational approach towards identifying contributing genes would be identifying the SNP polymorphism in the DNA sequences of the short-listed genes [11,12,13] between the parental cultivars of the QTL analysis. Particularly, the SNP variations that cause changes in the amino acid sequence might function differently, resulting in variations in the target phenotype. On the other hand, gene expressional difference is an obvious fact of causing phenotypic differences. As such, characterizing the expressional differences of the target genes across the parent and the contrasting population provides useful information to support the involvement of the candidate genes in the corresponding trait [14,15].

A recent compilation of yield QTLs in our laboratory from a total of 12 (year × treatment × location) trials of a Doubled Haploid (DH) population identified several QTL clusters. A significant QTL was detected on the long arm of chromosome 1B, covering the 15 centiMorgan (cM) genomic region. A significant contribution of this region was found to be associated with several important yield-related traits in multiple environmental conditions (Appendix A). The logarithm of the odd (LOD) and phenotypic variation explained (PVE) scores were also relatively higher than the threshold for several traits [16]. A phenotypic plasticity analysis of the mapping population demonstrated that the grain yield was strongly influenced by genetics, indicating the potential of the yield-related QTL to be used in the breeding program. However, the contributing gene(s) to the traits in the genomic region remained unexplored. It is necessary to explore the causal gene(s) in the selected region. Therefore, the above-mentioned QTL on the long arm of chromosome 1B has been selected for identifying the contributing gene(s) via an in silico approach and laboratory-based confirmation.

## 2. Results

### 2.1. Physical Location of the QTL in the Wheat Genome

The grain yield QTL was flanked by 90K SNP markers JD_c107_683 and RAC875_c55891_659. However, three additional markers, namely BS00063092_51, Tdurum_contig8158_269, and Tdurum_contig42852_667, were in the neighborhood genomic region. These five markers cover approximately 1.6 cM in the linkage map. Marker mapping on the latest wheat whole genome sequence assembly by using CLC Genomics Workbench identified the physical location of the region from 563,675,896 to 565,672,799 bp. Sixteen high-confidence (HC) genes and thirteen low-confidence (LC) genes (Appendix A) were identified in the region, which was intensively analyzed for the prioritization of the candidate genes in the downstream analysis.

### 2.2. Gene Functional Annotation

Functional annotation of the genes was obtained from the EnsamblPlant database (Biomart), and annotation descriptions of the genes were matched against IWGSC Refseq annotation v 1.1. Nucleotide and corresponding protein sequences were extracted from EnsamblPlant. The predicted protein functions were checked in several online databases like Uniprot (https://www.uniprot.org) (accessed on 8 August 2020), Pfam (https://pfam.xfam.org) (accessed on 8 August 2020), and EnsamblPlant (Table 1).

The annotation description of HC and LC genes and other information are presented in Appendix A. Available information from functional annotation indicates some proteins were parts of cellular components such as the plasma membrane, an integral part of the membrane, and intracellular anatomical structure. Molecular functions indicated HC genes were involved in ATP binding, GTP binding, GTPase activity, heme binding, protein binding, and oxidoreductase activity. The biological processes of the HC genes indicated their involvement in DNA replication, protein transport, photomorphogenesis, oxidoreductase activity, cell–cell signaling, chloroplast organization, and transcription regulation. The annotation description of most low-confidence genes is arabinogalactan peptide 14, which is involved in diverse developmental roles such as differentiation, cell–cell recognition, embryogenesis, and programmed cell death in Arabidopsis, but is absent in wheat. No other information is available for these genes. Therefore, based on the gene functional analysis results, the downstream analysis was focused on only the high-confidence genes.

### 2.3. Expressional Behaviour of the Genes (HC)

There are many genes in the wheat genome that do not have any expression and, hence, do not play any functional role in regulating any phenotype. The level of gene expression in biological samples is a powerful indicator of whether the gene has any functional role. A comprehensive expression pattern of the HC genes (Figure 1) was obtained from the Wheat Expression Browser (expVIP) (http://www.wheat-expression.com/) (accessed on 8 August 2020). Among the sixteen HC genes, only three (TraesCS1B02G336000, TraesCS1B02G336300, and TraesCS1B02G337800) showed expression over the threshold (expression unit: count (10), log_2_) [17], whereas the other thirteen genes were below the threshold or remain unexpressed (Appendix A). The highest expression of the selected genes appeared in the reproductive stage, followed by the vegetative and seedling stage. Expression levels were variable across the tissue types and developmental stages. The Wheat eFP Browser was used to check the individual gene expression, which resulted in a similar expression (Appendix A).

### 2.4. Protein–protein Interaction and Biochemical Pathway Analysis

Due to limited data available on wheat, the protein–protein interaction was predicted on rice orthologous proteins based on sequence similarity using the STRING database. The analysis demonstrated that the proteins of the selected genomic location have no interaction with each other. However, these proteins interact with several other proteins. While in many cases, the proteins of HC genes interact with proteins of unknown functions (Appendix A), some of the genes demonstrated interactions with known proteins. For example, rice homolog (OsJ_19113) of the wheat gene *ferredoxin* protein (TraesCS1B02G337100) was identified as interacting with the grain size (GS5) gene (Appendix A). KEGG pathway analysis of the interacting genes of this network showed involvement in cysteine and methionine metabolism, the biosynthesis of secondary metabolites, and the MAPK signaling pathway (Appendix A). Similarly, the sulfite reductase protein (TraesCS1B02G336300) also appeared to interact with known function genes such as nitrate reductase apoenzyme and adenylsulphate reductase magnesium-chelatase, serine acetyltransferase, etc. The majority of them are also involved in cysteine and methionine metabolism, sulfur metabolism, sulfur relay system, alanine, aspartate, and glutamate metabolism (Appendix A). Also, the gene *TPR* protein (TraesCS1B02G337400) was found interacting with several genes of known functions, such as wall-associated protein kinase, cell cycle regulatory genes, etc., alongside several genes with unknown functions. The analysis of the KEGG pathway revealed their involvement in sulfur metabolism and plant hormone signal transduction (Appendix A). On the other hand, the glutamine synthetase protein (TraesCS1B02G336800) interacted with several genes of known functions, for example, nitrogen-regulated genes and anaerobic respiration-controlling genes. The KEGG pathway analysis revealed their involvement in alanine, aspartate, glutamate metabolism, nitrogen metabolism, arginine and proline metabolism, arginine biosynthesis, amino sugar and nucleic acid sugar metabolism, pyrimidine metabolism, glyoxylate, and dicarboxylate metabolism (Appendix A).

### 2.5. Function Analysis of the HC Genes

On many occasions, the HC genes located in the QTL region were previously investigated in several experiments which reported their potential functional role. Thus, to relate the role of genes in influencing the corresponding trait for which this QTL was reported, the current study considered the reported function of those genes of great importance.

The analysis demonstrated that the orthologues of a few HC genes (proteins) of this study such as TraesCS1B02G336300, TraesCS1B02G336800, TraesCS1B02G337400, and TraesCS1B02G337100 were reported to be involved in the grain filling or yield contributing traits. For example, sulfite reductase [NADPH] hemoprotein beta-component (TraesCS1B02G336300) proteins were found to be involved in the chemical transformations for biogeochemical sulfur cycling [18] and protecting against sulfite toxicity [19]. It was also found that impairment in sulfite reductase led to early leaf senescence in tomatoes and decreased oxidative stress tolerance in *Arabidopsis* [20]. *TPR* (TraesCS1B02G337400) superfamily proteins were found to play a critical role in plant cell physiology, stress tolerance, and developmental processes [21]. Motifs required for abscisic acid responses and osmotic stress tolerance are present in such proteins. Plant responses to other hormones, for example, ethylene, cytokinin, gibberellin, and auxin in *Arabidopsis* also depend on the availability of these proteins [22]. *Ferredoxin*-like (TraesCS1B02G337100) enzymes were involved in electron transfer between *ferredoxin* and NADPH [23]. It acts as an electron carrier in photosynthesis and has positive effects on the photosynthesis rate and levels of carbohydrate metabolites [24]. It was also found to confer tolerance to heat stress and salinity [25]. Bifunctional glutamine synthetase adenylyltransferase/adenylyl-removing enzyme (TraesCS1B02G336800) regulates glutamine synthetase GlnA, which is a key enzyme in the ammonia assimilation process [26]. High cellular nitrogen levels reduce its activity. Again, low nitrogen levels increase its activity. The regulatory region of GlnE bound to the signal transduction protein PII (GlnB) indicates the cell’s nitrogen status [27].

On the other hand, investigation on previously reported gene function demonstrated that many of the genes located at the QTL region do not have the functional properties to influence the trait for which the QTL had been reported. For example, secretory carrier membrane proteins (SCAMPs) (TraesCS1B02G336000) are integral membrane proteins involved in membrane trafficking [28] and Populus wood formation [29]. WEB family protein (TraesCS1B02G336200) is involved in the chloroplast avoidance response under high-intensity blue light, which is involved in the salt stress tolerance in wheat [30]. ATP-dependent DNA helicase (TraesCS1B02G336400) has the function of removing the lagging strand in the replication forks [31], including repairing stalled replication forks. The U-box is a ubiquitin ligase activity-related protein domain [32] that plays an essential role in programmed cell death [33]. NBS-LRR disease resistance protein (TraesCS1B02G336700) provides resistance to different wheat diseases, such as leaf rust, powdery mildew, etc. [34,35].

### 2.6. Selecting the Candidate Genes for Lab-Based Investigation

Combining the findings of the above-mentioned in silico analyses, four out of sixteen HC genes (Table 2) appeared to have the potential to influence the target trait, which is grain yield. These genes play important roles in improving wheat grain yield directly or indirectly and were chosen for further investigation and functional confirmation.

### 2.7. Polymorphism of Candidate Genes between Parental Cultivars

The sequence alignments against the Chinese spring reference genome are presented in the following sections for every gene separately.

The sulfite reductase (TraesCS1B02G336300) gene was the largest among the four candidate genes and was sequenced into three parts. Details of the gene size, structure, and SNP variations are presented in Table 3 and Figure 2. The parental cultivar Westonia did not have any sequence variation with the reference cultivar Chinese Spring. However, it exhibited two SNP variations between the parental lines Westonia and Kauz in intron 5.

Diversity among wheat varieties (DAWN) showed the SNP variation for this gene among sixteen wheat cultivars, including Westonia (Appendix A). A total of seven SNP variations were observed among the 16 wheat cultivars, of which two were in the intron. The G/C SNP in the intron was matched with the Drysdale cultivar. The second SNP (T/C) was common in four cultivars, including Wyalkatchem, AC-Barrie, Baxter, and Volcani. However, SNP variations that resembled Westonia type or Kauz type were not observed among the sixteen wheat cultivars.

The glutamine synthetase (TraesCS1B02G336800) gene was the smallest in size. There was no sequence variation observed for the glutamine synthetase gene between the two parental cultivars and with the reference genome Chinese Spring. Also, the gene exhibited mostly conserved sequences in the DAWN database analysis. Out of sixteen cultivars, twelve did not have any SNP variation. There were a total of nine SNP variations between the four wheat cultivars, of which one was in the intron. Variations in the DNA sequences were mostly observed in Wyalkatchem and Baxter cultivars. Pastor and Xiaoyan-541 cultivars exhibited single SNP variation (Appendix A). The parental cultivars, Westonia and Kauz, exhibited conserved sequences for this gene.

The *ferredoxin*-like (TraesCS1B02G337100) gene exhibited two SNP variations in the second exon between the parental cultivars Westonia and Kauz (Figure 2). There was no variation in the DNA sequence between the reference genome and Kauz, whereas both SNPs were present in Westonia. The Kauz-type gene sequence was found conserved in seven wheat cultivars in the DAWN database analysis. The Westonia-type gene sequence was observed only in the RAC-875 cultivar (Appendix A). In Westonia, only the second SNP was associated with the amino acid change (Figure 2). There was a 7 bp insertion in the promoter region of Westonia (at −245 bp).

The *TPR* (TraesCS1B02G337400) protein genes have four SNP variations between the parental cultivars Westonia and Kauz. Two SNPs were present in the first exon of Westonia, and the other two were in the first intron of Kauz. There was a 2 bp deletion in Westonia in the first intron (Figure 2). The Westonia type gene sequence was conserved in the six wheat cultivars observed in the DAWN database analysis. The DAWN database exhibits huge SNP variations for this gene among the other ten cultivars. Nineteen SNP are observed mainly in ten cultivars, of which seven are in the intron. Single SNP variation is observed in the intron in RAC-875, Excalibur, and Chara (Appendix A). However, the Kauz-type gene sequence was not present in any cultivars. Due to the SNP variation, one amino acid change occurred during translation (Figure 2).

The sequence analysis of the selected genes revealed that only two genes, namely *ferredoxin* and *TPR*, exhibited polymorphism between the parental cultivars, indicating their involvement in the observed QTLs. Hence, expression analysis of the polymorphic genes was performed for these two genes only.

### 2.8. Association of Polymorphic Genes with the Yield

As the next step of functional prediction of the candidate genes with the target trait (grain yield), the expressional behavior of the two selected genes was found for the high-yielding and low-yielding DH lines at the seedling stage. Between the two parental lines, Westonia has better performances for the grain yield and yield contributing characteristics. Therefore, gene expression in the selected lines was compared with the better parent. For the *ferredoxin* protein gene, the high-yielding DH lines had higher values for the gene expression (M = 0.99, SD = 0.139) than the low-yielding DH lines (M = 0.568, SD = 0.141). The highest expression was obtained for the *ferredoxin* gene in the 01Y271-D02 line (1.10), which is statistically similar to the high-yielding parent Westonia (Figure 3). A t-test for independent samples (equal variance assumed) confirmed that the observed difference was statistically significant, t = 4.747, *p* = 0.001. Low-yielding lines had lower expression than the parental line, which was statistically significant t = 2.809, *p* = 0.04.

For the *TPR* gene, the highest expression was obtained in the high-yielding parent Westonia (Figure 4), which was similar to the high-yielding lines (*p* = 0.12) but statistically different from the low-yielding lines (*p* = 4.76 × 10^−5^). The high-yielding DH lines had higher values of gene expression (M = 0.99, SD = 0.044) than the low-yielding DH lines (M = 0.661, SD = 0.161). A t-test for independent samples (equal variance assumed) showed this difference was statistically significant, t = −7.526, *p* = 0.000.

The gene expression analysis revealed that the high-yielding DH lines and Westonia (better parent for grain yield) had a similar expression pattern for these genes. In comparison, the opposite gene expression pattern was observed in the low-yielding lines, which were statistically significant. The polymorphic status of these genes, as well as the gene expression pattern in the contrasting DH lines, provide more emphasis for being candidate genes for grain yield. 

## 3. Materials and Methods

### 3.1. Database Selection

On chromosome 1B, a significant QTL was observed for several yield-related traits [36]. Detailed information on the QTL is provided in Appendix A. Recently published wheat genome sequence and annotation, available at the International Wheat Genome Sequencing Consortium (IWGSC) sequence repository, were used as references. The 90K SNP markers were used to find the harboring genes in CLC Genomics Workbench 10 software package and online databases, including Biomart. Additional markers from publicly available genetic maps (https://wheat.pw.usda.gov/GG3/) (accessed on 8 August 2020) were also used to narrow down the selected region (Figure 5).

### 3.2. Basic Local Alignment Search Tool (BLAST)

The query gene sequences retrieved from IWGSC were examined using BLAST (https://blast.ncbi.nlm.nih.gov/) (accessed on 8 August 2020). BLAST was performed using gene and protein sequences based on bread wheat, durum wheat, rice, and *Arabidopsis*. Sequence similarity of the query sequence in related organisms was used to predict the gene function and other analyses.

### 3.3. Gene Expression Data Collection and Analysis

Wheat eFP browser (http://bar.utoronto.ca/efp_wheat/cgi-bin/efpWeb.cgi) (accessed on 8 August 2020) and Wheat Expression Browser (expVIP) (http://www.wheat-expression.com/) (accessed on 8 August 2020) were used to accumulate the expressional behavior of the gene. These databases stored the gene expression information of the previously published experiments, which cover a wide range of contexts [37,38]. Detailed information on the expressional behavior, such as time, tissue/organ, and expression level of all the listed genes, was compiled for a relative comparison. Only the expression levels above the threshold were considered for the gene selection [39].

### 3.4. Functional Annotation Analysis

Functional annotation of the selected genes was collected from publicly available databases, which include gene annotation of the IWGSC and Ensamblplant [40]. The predicted protein functions were checked in several online databases like Uniprot (https://www.uniprot.org) (accessed on 8 August 2020), Pfam (https://pfam.xfam.org) (accessed on 8 August 2020), and EnsamblPlant. The information considered from this analysis includes gene ontology, similarities, protein family descriptions, orthologues, and paralogues (Appendix A). All efforts were made to collect the gene functional information from the published works of literature and were considered to identify the functional properties of all genes of the QTL region.

### 3.5. SNP Data Collection

To identify the common alleles of the gene sequences, variant analysis was carried out using Plant Ensembl (https://plants.ensembl.org/) (accessed on 8 August 2020) and DAWN (Diversity Among Wheat geNomes) [41]. Additionally, results of gene variant analysis retrieved from RNAseq studies conducted within our laboratory were used to obtain information on the allelic variation of the selected genes.

### 3.6. Kyoto Encyclopedia of Genes and Genomes Pathway

To further understand the functional role of the genes of the QTL region toward the phenotype, gene pathway analysis was carried out using the Kyoto Encyclopedia of Genes and Genomes (KEGG) pathway platform. This platform collects pathway maps representing current molecular interaction, reaction, and relation networks for metabolism, cellular processes, and organismal systems. Detailed information on the selected genes was collected from this online database.

### 3.7. Protein–protein Interaction Prediction and Literature Support

Protein–protein interaction was analyzed using the STRING (https://string-db.org/) (accessed on 8 August 2020) platform based on the amino acid sequence [42,43]. The purpose was to understand the indirect influence of the genes on that phenotype (yield and yield contributing traits) through the gene network in model plant species.

### 3.8. Gene Amplification and Sequencing Analysis

Short-listed genes out of the screening with the in silico approach were sequenced from the two parental cultivars to know the existence of allelic variation, which is the prerequisite of contributing to the QTL. DNA of the parental lines (Westonia and Kauz) was extracted from the seed and leaf using a modified CTAB protocol [36]. Gene and chromosome-specific primers were designed for each gene. PCR amplification was performed in a thermocycler using touchdown PCR protocol. The reaction mixture contained 12.5 µL GoTaq Master mix, 2 µL of each primer (forward and reverse), 1 µL of DNA (200 ng) template, and nuclease-free water up to the final volume of 25 µL. The PCR product was then separated by electrophoresis and Promega 100 bp DNA ladder in 1% agarose gel in TAE buffer at 110 V for one hour, visualized under a UV transilluminator (Astral Scientific Pty Ltd, NSW, Australia), and photographed using the PolyDoc System. PCR products of the desired size were cut off from the gel and recovered using a gel purification kit (Promega Corporation, NSW, Australia). The sequencing reaction was performed using purified DNA fragments as a template. Then, the purification of the reaction product sequencing was performed [38]. The gene sequences were analyzed using Geneious 10 software (https://www.geneious.com/) (accessed on 8 August 2020). Accordingly, sequence alignment was performed using this software to identify the allelic variation between the parental cultivars.

### 3.9. RNA Extraction, cDNA Synthesis, and Quantitative PCR

Expression analysis of the selected genes was performed in the two pools of DH lines with contrasting phenotypes. Seeds of five high-yielding and five low-yielding DH lines were germinated on the Petri dishes. Total RNA was extracted from germinating seeds following the TRIzol (Invitrogen Australia Pty Ltd, NSW, Australia) method [43] and treated with DNaseI (Qiagen Pty Ltd, Victoria, Australia) according to the manufacturer’s specifications. RNA concentration was determined using a NanoDrop ND-1000 spectrophotometer (Thermo Scientific, Wilmington, DE, USA). For cDNA synthesis, one microgram of total RNA was reverse transcribed using Tetro™ cDNA synthesis kit (Meridian Bioscience, Cincinnati, OH, USA) in a 20 μL reaction. cDNA was then diluted into 100 ng/μL. Quantitative PCR was performed with an iTaq SYBR Green supermix (Bio-Rad, Munich, Germany) using specific primer pairs. Actin was used as the reference gene, and parental line Westonia was used as the control. The total volume for each reaction was 10 μL, containing 5 μL iTaq SYBR Green supermix, 2 μL cDNA, and 0.5 μL of each primer, adjusting the volume with distilled water. Gene quantification was performed using a Rotor-Gene Q (Corbett Life Science, Sydney, Australia). Thermal cycling conditions were set as follows: 95 °C for 3 min, followed by 30 cycles of 95 °C for 15 s, 60 °C for 15 s, and 72 °C for 30 s. Three biological replicates, each with three technical replicates, were analyzed, and the qPCR experiment was repeated to ascertain the reliability of the results. Linear data were normalized to the actin gene’s mean CT, and the relative expression ratio was calculated using the formula 2^−ΔΔCT^ [44]. Levene’s homogeneity test was performed on the data obtained from the qPCR analysis. An Independent sample T-test was conducted to investigate the mean difference in the expression level of the two groups of samples.

## 4. Discussions

In quantitative genetics, the association between genotype and phenotype is computed by a complicated statistical method where the holistic status of all genes and individual genes contributing to a specific phenotype remains as a black box. Advancement in molecular and quantitative techniques allows deciphering the black box of the polygenic control of quantitative traits. Recent progress in next-generation sequencing and whole-genome sequencing provide support in exploring the genetic architecture, distribution, and interaction of loci affecting the variations in characteristics [45,46,47,48,49].

Two approaches were followed for deciphering the black box, including genome-wide scanning and candidate gene approach, which both have pros and cons. Genome-wide scanning usually proceeds without having any prior knowledge of the target trait. The process is costly, time-consuming, and laborious. The candidate gene approach requires previous information on the target trait, which is a more effective and economical method for direct gene discovery. Lack of information is the bottleneck that might be overcome, as a fully annotated reference genome is now available [46,47]. Combining both approaches is an alternative way of candidate gene prioritization, which was followed in the current study.

This study followed a route (Figure 5) to prioritize candidate genes via a computational approach from a grain yield QTL in wheat. Functional analysis of all genes is not only impossible but also very expensive and time-consuming. The in silico process can substantially reduce the workload, time, and expense. QTLs are the statistical approach to link phenotypic data to genotypes. The QTLs need to be accurate and validated before any downstream activities [50,51]. Error in phenotyping and computational statistical analysis will produce a false QTL that may not influence the target trait. Predicting the gene function is also a challenge, as the protein is predicted based on sequence similarity, which might misguide the approach. The yield QTL used in this study harbored sixteen high-confidence genes in the chromosomal location. To extract the harboring genes, both the CLC Genomics Workbench software and the Biomart (EnsamblPlant) online database were used and matched with each other. For CLC Genomics Workbench, 90K SNP markers were mapped in the IWGSC RefSeq v1.1. The marker mapping provided the gene ID of the harboring genes along with the annotation description. Extraction of harboring genes using Biomart required the marker names on the selected chromosomes. It provided more information than CLC Genomics Workbench. Though the number of harboring genes varied a little bit between these two searches, the combination of both searches was used in the study. However, Biomart did not provide any information on the low confidence gene. Therefore, only CLC Genomics Workbench was used to extract low-confidence genes. Extraction of gene and protein sequences with CLC Genomics Workbench was difficult but easily doable with Biomart. An alternative way was to use the gene IDs to extract the gene sequence from the EnsamblPlant. The challenging part was matching the provided annotation descriptions. For this purpose, not only CLC Genomics Workbench, Biomart, and EnsamblPlant, but also Uniprot, Pfam, and all other available online databases were used based on sequence similarities in model and related plant species. All living organisms have a good number of similarities in several genes. However, prokaryotes were better studied than eukaryotes, and in eukaryotes, animals had more focus than plants. Although living organisms might have similarities for some genes, the ultimate expression and function might be different depending on the organisms. In most of the cases, the similarity of gene sequence indicated the gene annotation was reviewed in microorganisms but not for plants. Suggested unreviewed annotation descriptions for model and related plan species were used for such cases. The QTL region harbored thirteen low-confidence genes for which the bioinformatic analysis was not performed. The annotation description of these genes was ambiguous, and the predicted proteins were human-related, not plant-related. The level of gene expression was also very low. Therefore, the low-confidence genes were excluded from the intensive study of gene identification.

The function prediction of protein is also very useful based on the organism as well as additional data sources. Prior knowledge of the gene is essential for this purpose in the same species or model organism [45]. A comprehensive understanding of the protein function is required for the selected genes, which might be achieved from multiple data sources based on model organisms and neighboring species [52]. A basic local alignment search can also predict protein function on related organisms based on sequence similarity. Moreover, gene expression data can provide an excellent understanding of the gene function, though the data are limited to particular species and stage-specific. KEGG pathway analysis and protein–protein interaction data might be beneficial in predicting gene function and interaction with other genes. However, these data are very limited to specific organisms, though more and more information is added to different platforms regularly [50]. Therefore, a combination of all the approaches and supporting literature would be the most suitable way to predict gene function, as demonstrated in this study.

Functional confirmation of the candidate genes can be obtained by performing laboratory experiments. Gene overexpression and knock-down by producing transgenic lines may reveal the candidate gene function. Production of transgenic lines is time-consuming and costly and requires modern facilities and technical knowledge [38]. Sequencing of the candidate’s genes provides information on their gene structure and SNP variation. SNP is the simplest form of DNA sequence variation among individual plants and is responsible for the diversity among individuals [13]. SNPs can change the encoded amino acids or occur in the non-coding regions, influencing promoter activity (gene expression), mRNA stability, and subcellular localization of mRNAs. Therefore, identifying variations in gene expression and analyzing their effects may lead to a better understanding of their impact on gene function [13].

The gene expression analysis provides valuable models for studying complex and quantitative traits. The variations of gene expression patterns are directly responsible for the existing variations in the transcriptome and proteome. The rationale of the function-dependent strategy states that the differentially expressed genes are responsible for trait variation [46]. It is possible to extract the candidate gene responsible for the target trait from the pattern of gene expression profiles. Gene expression profiles are increasingly analyzed in searching for candidate genes. The analysis of protein–protein interaction, KEGG pathway, gene sequencing, and gene expression was discussed for individual candidate genes in the following heading.

### 4.1. Tetratricopeptide Repeat (TPR)-like Gene

Two nucleotide variation was observed between the sequences of the candidate gene *TPR*-like superfamily protein in the parental cultivars, indicating this gene can be segregated across the DH population and hence can contribute to the QTL. The Westonia allele of this gene was conserved in six wheat cultivars out of sixteen in the DAWN database analysis. The second SNP was associated with an amino acid change from alanine to glycine. The Kauz allele was not present among the sixteen cultivars. The single SNP variation, as well as changes in amino acid, was found to be associated with the variation of a specific trait in some studies. The C-A mutation in the GS3 gene played a critical role in seed size differences in rice [38]. Again, the C to A mutation was reported to be functionally associated with enhanced grain length in rice [39]. The A-T mutation in the GS5 gene was found to be significantly correlated with larger grain size and higher thousand kernels’ weight in wheat [41]. Accordingly, qPCR analysis exhibited a medium expression level for this gene in the high-yielding lines and low expression in the low-yielding lines. The candidate gene *TPR*-like superfamily protein is involved in the accumulation of photosystem I and functions as a PS1 biogenesis factor. This gene also showed a higher expression level in the root and a medium expression level in leaves and spikes observed at the Wheat eFP Browser (Appendix A). Photosystem I is an important component in chloroplasts and acts as a light-harvesting component. The activity of this vital component is directly related to the amount of photosynthate produced in photosynthesis. The level of expression of this gene in the leaves suggests its activity in producing food for the plant. In the STRING database, the rice homolog of this gene was found to interact with the cell cycle and cell wall-associated proteins (Appendix A) [42]. Interactions with cell cycle and cell wall-associated proteins indicate the role of the candidate gene in the fundamental process of growth and development. As such, the result suggested that the gene might also be involved in the higher yield trait in wheat. However, further analysis of the gene expression in different stages and transcriptional studies or gene editing experiments would be useful for further confirmation.

### 4.2. Ferredoxin-like Gene

Two nucleotide variations were observed between the *ferredoxin* gene sequences of the parents. A difference of glutamine to lysine at 615 bp was found for one SNP. The other SNP at 566 bp was not associated with a change in the protein sequence. In addition, there was a 7 bp insertion in the promoter region of Westonia. The qPCR analysis also exhibited a high expression level for this gene in high-yielding DH lines, which was resembling the parental line Westonia. On the other hand, significantly reduced gene expression was observed in the low-yielding DH lines. *Ferredoxin*s are iron–sulfur proteins that transfer electrons in a wide variety of metabolic reactions [43]. This gene is thought to be involved in numerous reactions. The literature supported its influence on thousand-grain weight by enhancing photosynthesis efficiency [40]. Moreover, in the publicly available databases, this gene showed a higher expression level in the inflorescence and a medium expression level in roots and spikes observed at the Wheat eFP Browser (Appendix A). The electron transport capacity and photosynthetic gas exchange rate were significantly higher in rice due to the constitutive expression of such a protein, which increased 1000-grain weight. A similar result is expected in the case of bread wheat. In the STRING database, the rice homolog of this gene was observed to interact with grain size gene GS5 (Appendix A) [42]. It also interacts with other genes, such as stress-related genes and genes involved in cation (Cu) loading in the xylem.

As a whole, the preliminary result suggested that the gene is actively involved in the improved yield and quality of high-yielding DH lines and parental line Westonia, hence contributing to the QTL. However, further analysis of the gene expression in different stages and gene overexpression or silencing experiments are recommended for further confirmation of the gene function.

### 4.3. Glutamine Synthetase Gene

Glutamate synthetase is a well-known gene for regulating glutamine synthetase, a key enzyme required to assimilate ammonia [42,43]. A higher expression level for this gene was obtained from the Wheat eFP Browser for different parts like roots, spikes, and glumes (Appendix A). STRING and KEGG pathway analysis revealed that the interacting proteins were involved in nitrogen metabolism, alanine, aspartate, and glutamate metabolism, arginine and proline metabolism, arginine biosynthesis, amino sugar and nucleic acid sugar metabolism, pyrimidine metabolism, glyoxylate, and dicarboxylate metabolism (Appendix A) [42]. The presence of this gene in the QTL region suggested it could be a contributing gene to the yielded QTL. However, laboratory analysis of the DNA sequencing for this gene between the parental cultivars Westonia and Kauz did not exhibit any SNP variation, indicating this gene has the same effect across the DH population and was not able to contribute to producing a QTL.

### 4.4. Sulphite Reductase Gene

In silico analyses showed that the sulfite reductase gene has a high expression level in different developmental stages at various tissues, such as root, flag leaf, and embryo (Appendix A). KEGG pathway analysis showed that this gene helps plants in the different physiological processes (Appendix A). The protein–protein interaction revealed its association with nitrate reductase apoenzyme 1, which plays an essential role in the nitrogen metabolism in plants (Appendix A) [42]. A high level of expression of this gene in roots, flag leaf, and spikes indicates its association with nutrient uptake, metabolism, and source-to-sink transport. Overall, gene function information and in silico analysis support that the sulfite reductase gene improves sulfur availability and increases the plant’s nitrogen status. As such, this gene could be the contributing gene to the targeted QTL. However, laboratory-based whole gene sequencing information demonstrated that there was no variation in the gene’s DNA sequence between the parental cultivar Westonia and Kauz, which suggests both of the parental cultivars carry the same allele of these genes. Thus, theoretically, it is impossible to segregate this gene across the DH population, which is essential for QTL identification. Thus, it can be concluded that this gene did not contribute to the studied QTL.

## 5. Conclusions

Prioritization of candidate genes via an in silico approach has appeared as a practical and time-worthy approach to QTL positional cloning. This approach provided a large amount of valuable information for selecting the candidate genes, which was further characterized by laboratory approaches. In silico analysis enabled the downsizing of the candidate gene numbers to 4 from 16 high-confidence genes located in the QTL region. Gene sequence polymorphism between two parental cultivars provided validation of the selected two genes (*ferredoxin*-like and *TPR* protein), while the other two appeared not to contribute to the QTL. qPCR-based gene expression analysis further demonstrated that both genes’ expression was related to the yield variation between the DH lines. Thus, this study showed that *ferredoxin* protein and *TPR* protein genes are potentially contributing genes for the targeted QTL. Gene over-expression or knock-down-based gene function confirmation would be useful for further validation.

## Figures and Tables

**Figure 1 plants-13-00026-f001:**
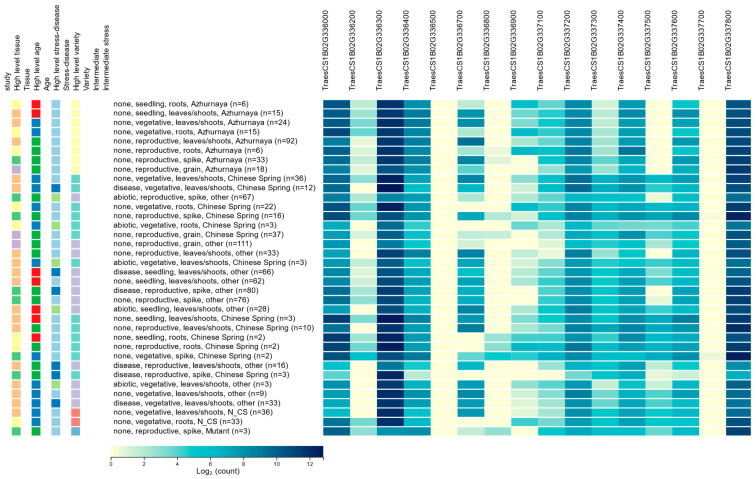
Level of expression of the HC genes obtained from Wheat Expression Browser (expVIP).

**Figure 2 plants-13-00026-f002:**
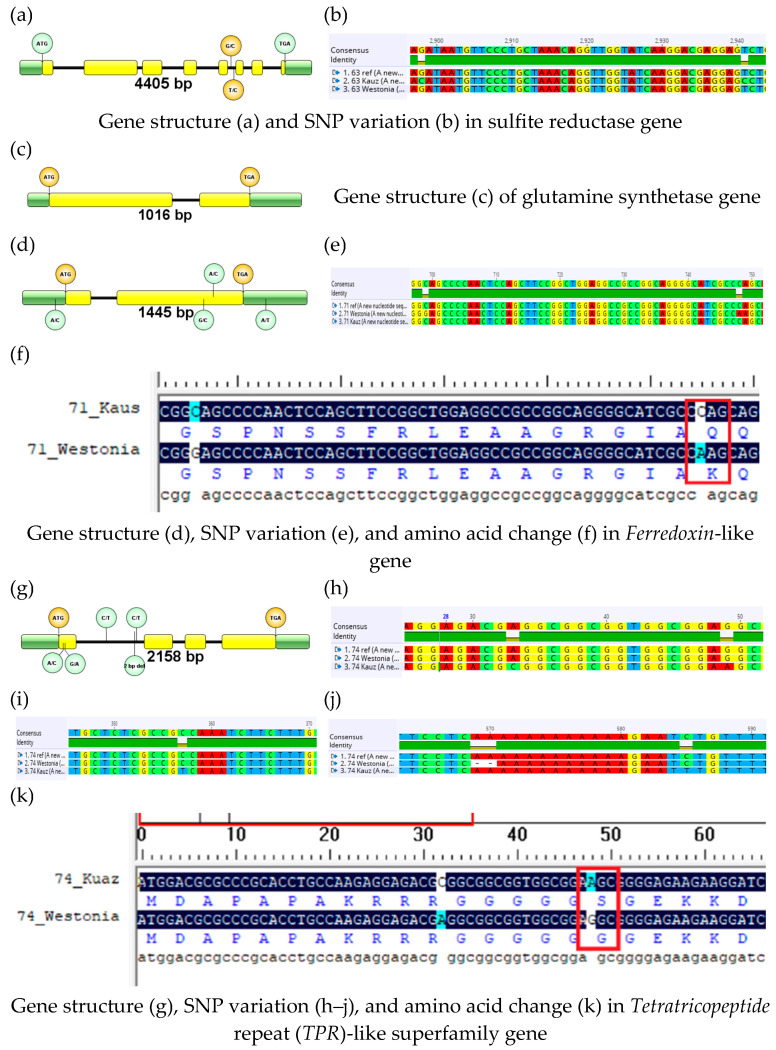
Gene structure and SNP detection followed by comparing the resultant amino acids for the selected genes of the parental cultivars Westonia and Kauz.

**Figure 3 plants-13-00026-f003:**
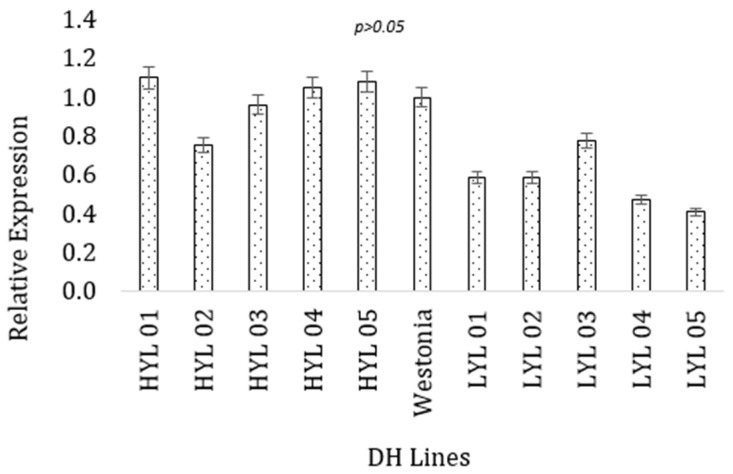
Expression of *ferredoxin* gene (TraesCS1B02G337100) in high and low-yielding lines. The vertical bar indicates the standard error.

**Figure 4 plants-13-00026-f004:**
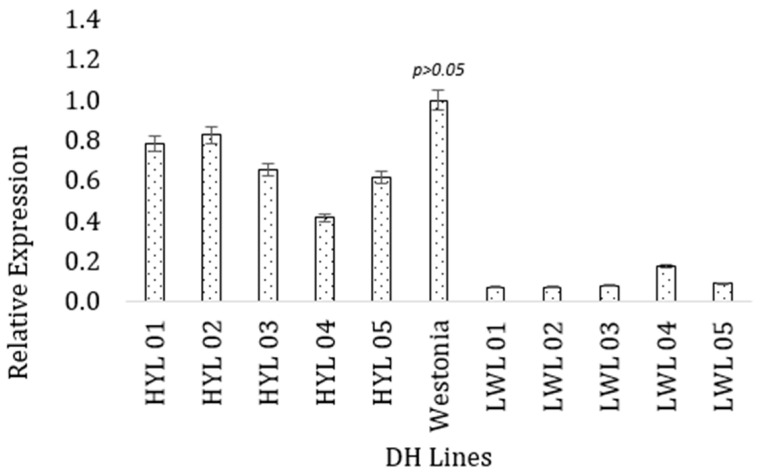
Expression of the *tetratricopeptide*-repeat gene (TraesCS1B02G337400) in high and low-yielding lines. The vertical bar indicates the standard error.

**Figure 5 plants-13-00026-f005:**
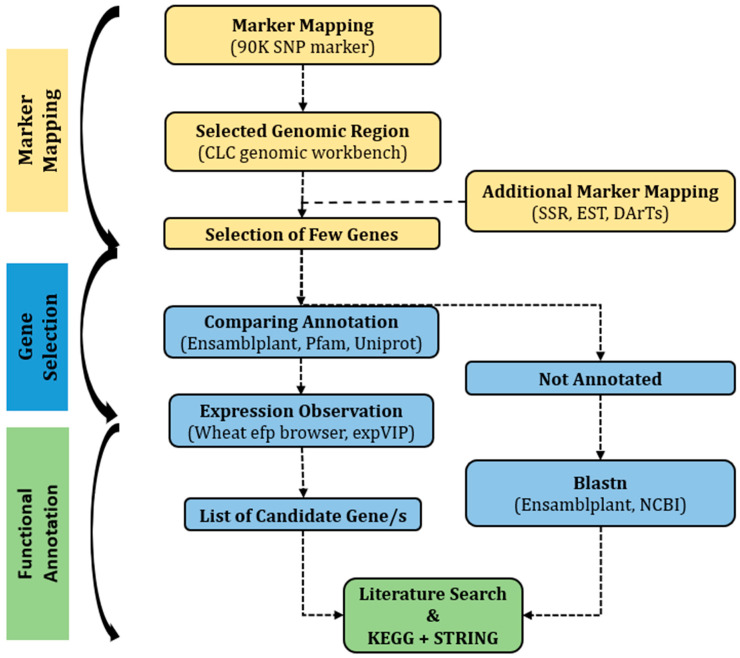
Flow diagram of the methodology for identifying candidate gene.

**Table 1 plants-13-00026-t001:** Annotation description of the HC genes.

Gene Ids	Annotation Description	Expression Level	Target Trait-Related	Scoring	Further Study
TraesCS1B02G336000	Secretory carrier-associated membrane protein	Low	NTT	2	N/R
TraesCS1B02G336200	WEB family protein (DUF827)	Low	NTT	2	N/R
TraesCS1B02G336300	Sulfite reductase [NADPH] hemoprotein beta-component	Medium/High	TT	1	Recommended
TraesCS1B02G336400	ATP-dependent DNA helicase Hel308	Low	NTT	3	N/R
TraesCS1B02G336500	RING/U-box superfamily protein	No	NTT	3	N/R
TraesCS1B02G336700	NBS-LRR disease-resistance protein	Low	DR/NTT	3	N/R
TraesCS1B02G336800	Bifunctional glutamine synthetase adenylyltransferase/adenylyl-removing enzyme	Low	TT	1	Recommended
TraesCS1B02G336900	Clavata3/ESR (CLE) gene family member	Low	NTT	3	N/R
TraesCS1B02G337100	Ferredoxin-like	Low	TT	1	Recommended
TraesCS1B02G337200	Rac-like GTP-binding protein	Low	DR/NTT	3	N/R
TraesCS1B02G337300	LRR receptor-like serine/threonine-protein kinase FLS2	Low	DR/NTT	3	N/R
TraesCS1B02G337400	*Tetratricopetide* repeat (*TPR*)-like superfamily protein	Medium/High	TT	1	Recommended
TraesCS1B02G337500	Actin cross-linking protein, putative	No	NTT	3	N/R
TraesCS1B02G337600	ARM repeat superfamily protein	Low	NTT	3	N/R
TraesCS1B02G337700	Actin cross-linking protein, putative	No	NTT	3	N/R
TraesCS1B02G337800	Transcription elongation factor Spt5-like protein	Medium	NTT	2	N/R

NTT: non-target trait, DR: disease-related, TT: target trait, and N/R: not recommended. The details are shown in Appendix A. Scoring: 1: mostly contributing to the target trait directly, 2: may have some influence on the target trait indirectly, and 3: not related to the target trait.

**Table 2 plants-13-00026-t002:** List of the four candidate genes.

Serial No.	Feature ID
Candidate gene 1	Sulfite reductase [NADPH] hemoprotein beta-component (TraesCS1B02G336300)
Candidate gene 2	Bifunctional glutamine synthetase adenylyltransferase/adenylyl-removing enzyme (TraesCS1B02G336800)
Candidate gene 3	*Ferredoxin*-like (TraesCS1B02G337100)
Candidate gene 4	*Tetratricopetide* repeat (*TPR*)-like superfamily protein (TraesCS1B02G337400)

**Table 3 plants-13-00026-t003:** Characterization of the candidate genes.

**Gene Id**	**Genomic Position**	**Gene Size** **(bp)**	**Protein Size (aa)**	**Number of Exons**	**Number of Introns**	**SNP Variation in Exon**	**Total SNPs**	**Amino Acid Change**	**Protein Structure Change**
TraesCS1B02G336300	563,031,942–563,027,538	4405	636	8	7	-	2	-	-
TraesCS1B02G336800	563,787,308–563,788,323	1016	213	2	1	-	-	-	-
TraesCS1B02G337100	564,161,126–564,162,570	1445	255	2	1	2	2	Yes	No
TraesCS1B02G337400	564,907,284–564,909,441	2158	301	4	3	2	4	Yes	No

## Data Availability

The data presented in this study are available on request from the corresponding author. The data are not publicly available due to privacy.

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
