# Peer review of "Identifying and Characterizing Candidate Genes Contributing to a Grain Yield QTL in Wheat"

_plants, 2023, doi:10.3390/plants13010026_

Round 1

Reviewer 1 Report

1. In reference 4, the DOI does not work.

2. In the materials and methods section (Zhao et. al., 2019) should be put into appropriate form, like the other references.

3. I think references 17, 18 and 23 are not relevant.

4. Missing references after 55.

Reviewer 2 Report

The article is devoted to prioritization of candidate genes via an in-silico approach. This approach provided a large amount of valuable information for selecting the candidate genes, which was further characterized by laboratory approaches. The text of the manuscript is well written, the introduction provide sufficient background and include all relevant references, the methods are adequately described, the results are clearly presented. The design of the figures could be improved.
Some minor comments:
- two times “marker-assisted selection (MAS)”
- bread wheat (Triticum aestivum) -> Triticum aestivum L.
- at the end of the introduction, the purpose of the article should be more clearly stated.

I did not notice significant errors in English.

Reviewer 3 Report

Manuscript is well organized and clearly presented. Only issue that I have is the references require attention to italicize species names e.g.

3. An, D., Su, J., Liu, Q., Zhu, Y., Tong, Y., Li, J., Jing, R., Li, B., & Li, Z. (2006). Mapping QTLs for nitrogen uptake in relation to the early growth of wheat (Triticum aestivum L.).  Triticum aestivum 
